# Heparin Protects Severe Acute Pancreatitis by Inhibiting HMGB-1 Active Secretion from Macrophages

**DOI:** 10.3390/polym14122470

**Published:** 2022-06-17

**Authors:** Jing Yang, Xujiao Tang, Qingqing Wu, Panpan Ren, Yishu Yan, Wei Liu, Chun Pan

**Affiliations:** 1School of Life Sciences and Health Engineering, Jiangnan University, Wuxi 214122, China; yangjing@jiangnan.edu.cn (J.Y.); 6201507011@stu.jiangnan.edu.cn (X.T.); 6191502015@stu.jiangnan.edu.cn (Q.W.); 6201507007@stu.jiangnan.edu.cn (P.R.); 2Jiangsu Key Laboratory of Druggability of Biopharmaceuticals, State Key Laboratory of Natural Medicines, School of Life Science and Technology, China Pharmaceutical University, Nanjing 210009, China; liuwei@cpu.edu.cn; 3Department of Critical Care Medicine, Zhongda Hospital, Southeast University, Nanjing 210009, China; panchun1982@gmail.com

**Keywords:** heparin, severe acute pancreatitis, HMGB-1, macrophage

## Abstract

Heparin has shown benefits in severe acute pancreatitis (SAP) therapy, but the underlying mechanisms were unknown. Extracellular high-mobility group protein-1 (HMGB-1) has been regarded as a central mediator contributing to inflammation exacerbation and disease aggravation. We hypothesized heparin attenuated the disease by targeting HMGB-1-related pathways. In the present study, the possible therapeutic roles of heparin and its non-anticoagulant derivatives, 6-O-desulfulted heparin and N-acylated-heparin, were determined on mouse models induced by “Two-Hit” of L-arginine. The compounds exhibited potent efficiency by substantially decreasing the pancreatic necrosis, macrophage infiltration, and serum inflammatory cytokine (IL-6 and TNF-α) concentration. Moreover, they greatly reduced the rapidly increasing extracellular HMGB-1 levels in the L-arginine injured pancreases. As a result, multiple organ failure and mortality of the mice were inhibited. Furthermore, the drugs were incubated with the RAW264.7 cells activated with damaged pancreatic tissue of SAP mice in vitro. They were found to inhibit HMGB-1 transfer from the nucleus to the plasma, a critical step during HMGB-1 active secretion from macrophages. The results were carefully re-examined with a caerulein and LPS induced mouse model, and similar results were found. The paper demonstrated heparin alleviated SAP independent of the anti-coagulant functions. Therefore, non-anticoagulant heparin derivatives might become promising approaches to treat patients suffering from SAP.

## 1. Introduction

Acute necrotizing pancreatitis is associated with the high rates of morbidity and mortality of severe acute pancreatitis (SAP). The disease often develops in a cascade of inflammatory reactions, sepsis, and multiple organ dysfunction, and is associated with a mortality rate of 8~39% after a “second hit” [1]. There has been a substantial evolution and development of intervention strategies in recent years including open surgery and minimally invasive surgical and endoscopic step-up approaches [2]. Nevertheless, these invasive approaches lead to high rates of adverse events and death without curative results [3]. In contrast, early pharmacological intervention to prevent or reduce pancreatic necrosis has become the fundamental strategy for the disease therapy. However, there are currently no licensed drugs available in clinics [4]; treatment is still limited to supportive care and management of complications [5].

During the development of SAP, extracellular high-mobility group protein-1 (HMGB-1) was recognized as an important predictor of the disease [6]. It is usually produced in the advanced stage of inflammation. The level in pancreatic tissues rapidly increases in experimental SAP models, leading to acinar necrosis, intestinal mucosal damage and dysbacteriosis, multiple organ failure, and death [7]. In fact, a high level of extracellular HMGB-1 is always associated with poor prognosis of patients [8]. The intervention of HMGB-1 secretion appears to be a promising alternative option for SAP therapy.

Heparin, an important group of antithrombin agents, has shown benefits during SAP treatment; they were able to reduce the incidence of pancreatic encephalopathy and improve the survival percentage both as monotherapy and in combination with other drugs [9,10,11]. Some research has attributed this efficacy to the improvement of local circulation and reduction of plasma D-dimer level [12].

However, heparin is also regarded as an important anti-inflammatory agent with multiple molecular targeting properties though its high affinity with a variety of chemokines, cytokines, and growth factors [13]. It represents a safe intervention for the treatment of various other inflammatory disorders [14]. To date, the underlying mechanism of how heparin exerts benefits in the inflamed pancreas–and more importantly, avoids multiple organ failure–is still controversial.

In this paper, we hypothesize that heparin alleviated SAP by targeting the HMGB-1 related pathway. We compared the efficacy of heparin, 6-O-desulfulted heparin (6-desO H), and N-acylated-heparin (N-actyl H) on an SAP mouse model with a “Two-hit” of L-arginine. Then, the related signaling pathway was investigated in vitro with activated RAW264.7 cell lines. Finally, the results were re-examined with a caerulein and LPS induced SAP mouse model. We concluded that heparinoid alleviated SAP by inhibiting HMGB-1 secretion from the pancreatic macrophages (Figure 1).

## 2. Materials and Methods

### 2.1. Materials

L-Arginine was purchased from Bomei Pharmatech Ltd. (Hefei, China). Mouse IL-6 and TNF-α enzyme-linked immunosorbent assay (ELISA) kits were from Proteintech (Wuhan, China). FITC labeled Dextran (Mr 100,000) was purchased from Xi’an Ruixi Biological Technology Co., Ltd. (Xi’an, China). Mouse HMGB-1 ELISA kits were purchased from Sangon Biotech (Shanghai, China). The Hematoxylin and Eosin (H&E) Staining Kit was purchased from Abcam (Kamblidge, UK). The anti-HMGB-1 rabbit polyclonal anti-body was purchased from Sangon Biotech. (Shanghai, China). Lipase activity and α-amylase activity assay kits were from Jiancheng Biotech (Nanjing, China). Caerulein was purchased from Nanjing Peptide (Nanjing, China). Lipopolysaccharide (LPS) was purchased from Aladdin (Shanghai, China).

### 2.2. Animal Experiment Design

#### 2.2.1. The L-Arginine Induced SAP Model

The Animal Research Committee of Jiangnan University approved all the animal experiments. Male ICR mice (8 weeks) were purchased from Cavens Laboratory Animal CO.LTD (Changzhou, China).

For the L-arginine induced SAP model, 37 mice were divided into the following five groups: the SAP group (n = 16), the heparin group (n = 14), the 6-desO H group (n = 14), the N-acetyl H group (n = 14), and the control group (n = 5). They had free access to food and water during the whole project.

The mouse model was set up according to the published paper [15]. L-arginine solution (8% *w*/*w*, pH = 7.4) was incubated in ice-cold water and injected intraperitoneally into the mice twice at an interval of one hour, with each dosage of 4 g/kg (approximately 1.3 mL). The mice were allowed to recover for 72 h and were then repeatedly subjected to L-arginine injection after the first-round operation. The group that received the identical PBS injections were used as control group.

The heparin, 6-desO H, and N-acetyl H groups were intraperitoneally administered with drugs (0.5 mg/kg) 1 h after the first L-arginine injection and received drug treatment every 24 h. Meanwhile, the SAP group and the control group received 0.1 mL of PBS buffer.

Blood was drawn from the mice, and they were sacrificed 24 h after the last time of L-Arginine induction. The pancreatic, lung, and intestinal tissues were taken out for further analysis. However, eight mice of the SAP and drug-given groups were allowed to remain in the cage for 80 h, which was used to calculate the survival rate in response to L-arginine challenge.

#### 2.2.2. The Caerulein and LPS-Induced SAP Mouse Model

Twenty-five ICR mice were used for the caerulein and LPS -induced SAP model setting. They were divided into the following five groups: the SAP group (n = 5), the heparin group (n = 5), the 6-desO H group (n = 5), the acetyl H group (n = 5), and the control group (n = 5). The mice were intraperitoneally injected with caerulein (200 µg/kg) each hour during a periold of 10 h, followed by LPS (5 mg/kg) administration 1 h after the last caerulein injection. They were free to access food and water during the whole project. The group that received the identical PBS injections were used as control group.

The heparin and its derivative groups were intraperitoneally administered with drugs (0.5 mg/kg) 1 h after the first caerulein injection. Meanwhile, the SAP group and the control group received 0.1 mL of PBS buffer. Blood was drawn from the mice, and they were sacrificed 12 h after LPS administration. The pancreatic tissues were taken out for further analysis.

### 2.3. H&E Staining and TUNEL Staining

Fresh tissues were fixed in 4% paraformaldehyde and embedded with paraffin. H&E staining was performed according to the published papers [16]. The pathological slices were scanned and recorded on a Pannoramic SCAN II instrument (3D Histech, Budapest, *Hungary*). Two investigators who were blind to the experimental treatment scored the pancreatic necrosis degree according to the published scoring standards [15]. TUNEL staining was performed according to the manufacturer’s instructions (KeyGEN Biotech, Nanjing, China). TUNEL-positive cells were observed in the fluorescence microscope.

### 2.4. Immunohistochemistry Staining (IHC) and Immunofluorescence Staining (IF)

The fresh pancreas was fixed, sliced, dewaxed, rehydrated, underwent antigen retrieval, were blocked with 5% BSA, and stained with the antibody of HMGB-1 or CD68. The IHC procedures were performed according to the standardized process [17]. Finally, the images were scanned and recorded on a Pannoramic SCAN II instrument (3D Histech, Budapest, *Hungary*).

### 2.5. Serum α-Amylase and Lipase Assays

The blood from mice was collected and centrifuged (4000 rpm/min, 10 min, 4 °C) to obtain the supernatant. Serum activity of α-Amylase and Lipase were detected using assay kits. All the kits were used according to the manufacturer’s instructions. Briefly, the lipase catalyzed the hydrolysis of oil esters into fatty acids, and the production rate was determined by the copper soap method based on the absorbance of at 715 nm. α-Amylase cleaved the substrate in serum to produce smaller fragments with the ability to form chromophore after I_2_ had been added. The product could then be measured based on the absorption at 660 nm.

### 2.6. Enzyme-Linked Immunosorbent Assay (ELISA)

Serum TNF-α, IL-6 and HMGB-1 were measured by ELISA kits (Proteintech, Wuhan, China) according to the manufacturer’s protocols.

### 2.7. Quantification of Intestinal Permeability

Intestinal permeability was measured as described previously [18]. Six hours after the induction of pancreatitis, 5 cm of terminal ileum and right-side colon were removed. The intestinal lumen was gently washed, and one side of the intestine was ligated. Next, FITC-Dextran (MW, 100,000 Da, 40 mg/mL, 200 μL) was perfused to the intestinal lumen, and then the other side was ligated. The intestinal pouch was shaken gently in 20 mL of saline at 37 °C for 60 min. The permeability of the intestinal wall was evaluated ex vivo by measuring the leaked amount of FITC-dextran outside of the intestinal pouch.

### 2.8. Cell Culture and Treatment

RAW264.7 cells were purchased from Cobioer Biosciences Co., Ltd. (Nanjing, China), cultured in DMEM with 10% FBS and 1% Penicillin-Streptomycin solution and maintained in a humidified atmosphere at 37 °C with 5% CO_2_. The cells were pretreated with heparin and its derivatives (50 μg/mL) before stimulation with supernatant of the pancreatic tissue from the “Two-Hit” model mice for 2 h (5 μL in 1 mL culture media).

### 2.9. Immunofluorescence Analysis

RAW 264.7 cells were plated on coverslips and cultured overnight. Two hours after treatment with pancreatic tissue supernatant and drugs, the cells were fixed, permeabilized, and blocked with 5% BSA. Cells were then incubated with mouse anti–HMGB-1 primary antibody (1:200 dilution) overnight at 4 °C, followed by incubation with Alexa Fluor 488 conjugated goat anti-rabbit secondary antibody for 45 min. Nuclei were counterstained with DAPI. Images were recorded using a Carl Zeiss LSM880 microscope (USA).

### 2.10. Western Blot

Two hours after treatment with pancreatic tissue supernatant and drugs, the cells were harvested by centrifuge (1500 rpm/min for 5 min). Total cellular proteins were isolated. Western blot analysis was performed using specific primary antibodies and horseradish peroxidase–conjugated secondary antibodies, as described previously [19]. Immunoreactive bands were detected using electrochem iluminescence reagents. The density of immunoblots was analyzed using Image Lab software.

### 2.11. Statistics

Data were expressed as mean ± SEM. Difference among multiple groups was assessed using one-way variance followed by the Bonferroni’s Multiple Comparison Test. The results were considered statistically significant at *p* < 0.05. All the analyses were conducted using GraphPad Prism 8 software.

## 3. Results

### 3.1. Protective Effect of Heparin against L-Arginine Induced SAP

To exclude the effect of the anti-coagulant activity of heparin, we prepared 6-desO H and N-acetyl H as previously reported. The molecular weight and distribution were determined with the same method (Appendix A) [20]. The ^1^H-NMR and ^13^C-NMR spectra were recorded and compared with the published paper (Appendix A) [21]. Then, we first examined their effect on the SAP mouse model induced by “Two Hits” of L-arginine. As shown in Figure 2A, extensive acinar necrosis occurred, with mast inflammatory cells infiltrated in the SAP group, indicating the pancreas was greatly impacted under two round attacks of L-arginine. On the other hand, the histological changes were substantially reduced in the heparin, 6-desO H, and N-acetyl H groups. Although the acinar cells had swollen compared with the control group, their tissue histoarchitectures were relatively complete compared with the SAP group. In addition, the acinar cell necrosis (Figure 2E), interlobular space (Figure 2D), and inflammatory cell infiltration (Figure 2F) was significantly reduced after drug treatment.

Consistent with these results, the TUNEL positive cells were found extensively in the pancreatic tissue of the SAP group, demonstrating an extremely high rate of cell apoptosis (Figure 2B), but were reduced substantially in response to the drug treatment. Meanwhile, the α-amylase and lipase activity in the serum increased to some degree in the SAP group and descended to near normal level after treatment with the drugs (Figure 2G,H).

The biological functions of heparin are believed to be dependent on the interaction with key proteins. The charge density of heparin contributes largely to the affinity between the bio-macromolecules. The interaction for N-acetyl H and 6-desO H are weakened due to the reduced charge density after partially desulfation [22]. As a result, heparin exerted a better effect compared with the N-acetyl H and 6-desO H groups. These results primarily indicated that all the heparin derivatives reduced the cell necrosis of SAP.

### 3.2. Heparin Prevented HMGB-1 Secretion and Multiple Organ Failure

To further evaluate the anti-inflammatory effects of heparin, we determined the inflammatory cells infiltrated in the pancreatic tissue. We found only the population of the macrophages substantially elevated in the pancreas of the SAP models. Heparin and its derivatives greatly reduced the number of macrophages in the pancreas (Figure 3A). The pro-inflammatory cytokine levels in the serum including TNF-α and IL-6 were also increased in the SAP group, but decreased in the drug group (Figure 3B,C).

More important, extracellular HMGB-1, the biomarker of advanced stage of SAP, increased about six times compared with the control group (Figure 3D). The extracellular HMGB-1 was extensively observed in the pancreas by immunofluorescence (Figure 3E), indicating that most of the pancreas tissue was damaged. However, HMGB-1 decreased to the normal level after the heparin administration both in the serum and in the pancreas. The 6-desO H and N-acetyl H group displayed similar effects.

HMGB-1 drives the inflammatory cascade and the following multiple organ failure. As such, we also compared the intestinal injury of the SAP and the drug groups. Indeed, the epithelium layer of the intestinal tissue was broken into small pieces, and the base membrane became very thin and even shed off in the SAP group (Figure 4A). After drug administration, the height of the epithelium layer increased, and the mechanical barrier integrity was much higher than in the SAP group. We also evaluated the leakage of the intestine by measuring the escaped FITC-Dextran outside the intestine. The intestine became highly permeable after the L-arginine attack (Figure 4C). Heparin derivative groups, especially the heparin and 6-desO H group, significantly reduced the leakage amount.

Acute lung injury (ALI) is one of the most serious complications during SAP development [23]. In the present study, ALI occurred in the SAP group along with significant alveolar collapse according to histological observation. The whole lung had a permeability of inflammatory and red blood cells. Heparin derivative groups alleviated the edema, infiltrations of inflammatory cells, and high permeability (Figure 4B). As a result, they reduced the morality of the mice in a period of 80 h (SAP 100% at 48 h), with N-acetyl H displaying the best efficacy (50% at 80 h) (Figure 4D). These results strongly supported that heparin provided protection effects during the SAP progression through the inhibition of extracellular HMGB-1 production.

To investigate whether heparin inhibited HMGB-1 active secretion from macrophages, the drugs were incubated with the activated RAW264.7 cells. HMGB-1 usually binds with DNA in the nuclei of the cell, transfers to the cell cytosol, and finally exits the cell under stress. The immunological fluorescence assay found the HMGB-1 concentrated in the nuclear zone of the normal control cell, whereas it shifted and diffused in the cytoplasm in the SAP group (Figure 5A). The allotopia was greatly reduced in response to the drugs (50 μg/mL), with more HMGB-1 merged with DAPI in the nuclear zone. In addition, the Western-blot against HMGB-1 also confirmed that the drugs increased HMGB-1 in the nuclear zone in a concentration-dependent manner, indicating they prevented HMGB-1 transferring to the cytoplasm (Figure 5B–D). The results strongly demonstrated that heparin blocks the active secretion of HMGB-1 from macrophages.

### 3.3. Confirming the Effect of Heparin against Caerulein and LPS Induced Severe Acute Pancreatitis

We then re-evaluated these effects on the most widely recognized SAP model induced by cerulein and LPS [24]. The histological analysis found acute pancreatitis occurred immediately (Figure 6A). Similarly, the drug administration alleviated the acinar cell swelling and necrosis, inflammation, and reduced the α-amylase (Figure 6B) and lipase activity (Figure 6C). They all displayed anti-inflammatory effects by reducing TNF-α (Figure 6D) but not the IL-6 concentrations (Figure 6E) in the serum. The drugs also strongly inhibited macrophage infiltration (Figure 6F) and decreased extracellular HMGB-1 levels in the pancreas through immunohistochemical observation (Figure 6G). The results were completely consistent with the data obtained on the L-arginine induced SAP models.

## 4. Discussions

With the continued increasing incidence of SAP, the drug development for the disease therapy is urgent. Recently, heparin has shown to attenuate the pancreatic damages. The involved pathogenesis is unknown, and only attributed to the prevention of splanchnic thrombosis [25]. However, heparin has also shown beneficial effects in various inflammatory disorders, independent of the anticoagulant properties. To further investigate the potential therapeutic value of heparin, it is necessary to elucidate the mechanisms by which heparin offer therapeutic potential. 

In this paper, we prepared the non-anticoagulant form of heparin and evaluated its effects on the mouse model induced by “Two-hits” of L-arginine. All the drugs substantially alleviated the pancreatic necrosis and strongly reduced the macrophage recruitment and the cytokine levels. As pancreatic necrosis was considered an absolute indication of poor prognosis of the patients, this was the key cellular processes for heparin to control the pathogenesis of SAP [26].

HMGB-1 is a constitutive protein that stabilizes nucleosomes in almost all cell lines in normal tissues. It is passively released during cell necrosis, or actively released from immune cells stimulated by pro-inflammatory mediators [27,28]. The intracellular and extracellular HMGB-1 played distinct roles as SAP developed. Intracellular HMGB1 appeared to prevent nuclear catastrophe and release of inflammatory nucleosomes to block inflammation [29]. By contrast, extracellular HMGB-1 is recognized by the innate immune system as a “necrotic marker” to signal tissue damage, closely relating to the following multi-organ dysfunction and high mortality.

Some researchers have reported that heparin reduced the amplification of several other inflammatory responses induced by HMGB-1 by interacting with HMGB-1 directly [30], or interfering with the interaction of HMGB-1 with its cell surface receptors [31]. We speculated that heparin also alleviate SAP through HMGB-1related pathways.

Indeed, extracellular HMGB-1 was found in a huge amount in the pancreas of the SAP group, indicating the disease progressed into serious conditions. The drug administration substantially reduced the extracellular HMGB-1 levels. Consequently, the multiple organ failure (e.g., ALI) had been blocked and the mortality was decreased.

Furthermore, as macrophages were regarded as the main source of extracellular HMGB-1, we investigated whether heparin inhibited HMGB-1 secretion from macrophages [32]. We incubated RAW264.7 cells, which were activated with damaged pancreatic tissue of SAP mice, with heparin and its non-anticoagulant derivatives in vitro. We found that all the drugs inhibited HMGB-1 from shifting from the nucleus to the cytoplasm, the initial step during HMGB-1 active secretion. As such, we concluded that they inhibited HMGB-1 active secretion from the macrophage.

Then, the conclusion was carefully re-examined with a caerulein and LPS induced SAP model. Similarly, the drugs alleviated reduced the acinar cell swollen and necrosis in pancreas and reduced the serum TNF-α concentrations due to decreased HMGB-1 levels in the pancreas tissue. The results were consistent with the ones obtained on L-arginine induced SAP.

## 5. Conclusions

In summary, we evaluated the therapeutic effects of heparin and its non-anticoagulant derivatives on SAP mouse models and investigated the related mechanisms. We found they effectively alleviate the pancreatic necrosis and high inflammatory state by blocking HMGB-1 active secretion from macrophages, demonstrating that heparin exerted the effects independent of the anti-coagulant function. Given the safe profile of the non-anticoagulant heparin derivatives, they might replace heparin and become promising approaches that treat patients suffering from SAP.

## Figures and Tables

**Figure 1 polymers-14-02470-f001:**
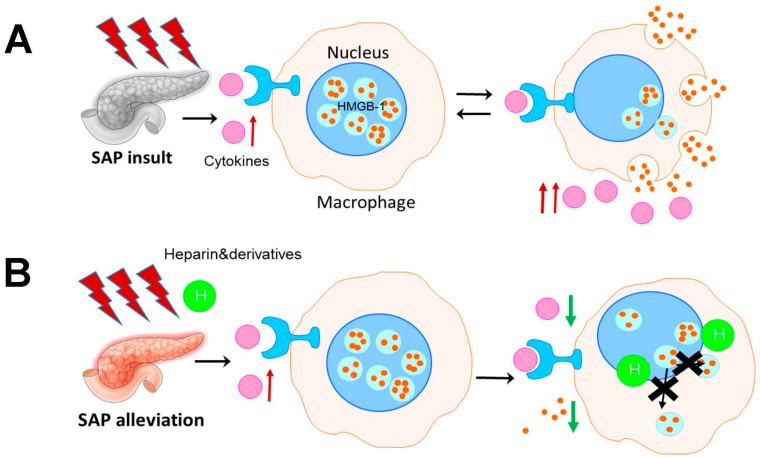
The mechanism through which heparin and derivatives exert effects on SAP. (**A**) Extracellular HMGB-1 secreted from activated macrophages drive the inflammatory exacerbation and lead to acinar necrosis in SAP. (**B**) Heparin and non-anticoagulant derivatives alleviated the disease by blocking the allotopia of HMGB-1 from the nucleus of macrophages, thereby inhibiting the secretion.

**Figure 2 polymers-14-02470-f002:**
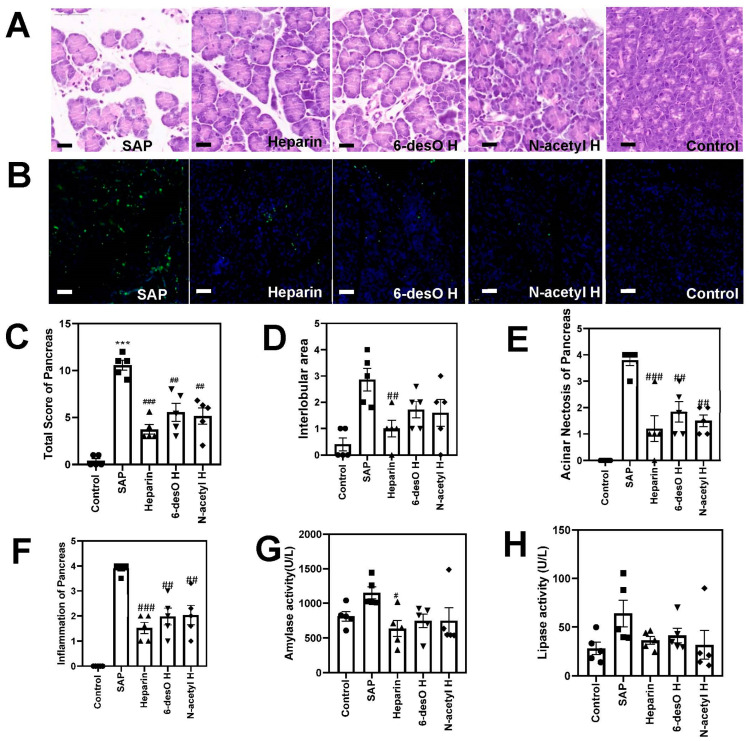
The effect comparison between heparin and its derivatives on “Two-Hits” of L-arginine induced mouse SAP. (**A**) The representative images of the pancreas H&E staining. (**B**) The TUNEL staining of the pancreas tissue. (**C**) The total histological score of the pancreas damage degree analysis. (**D**–**F**) The histological analysis of the pancreatic damage in terms of interlobular area (**D**), acinar necrosis (**E**), and (**F**) inflammation. (**G**) The serum α-amylase activity analysis. (**H**) The serum lipase activity analysis. Scale Bars = 20 μm. (# *p* < 0.05, ## *p* < 0.01, ### *p* < 0.005, SAP vs. drug administration groups; *** *p* < 0.005, SAP vs. control groups).

**Figure 3 polymers-14-02470-f003:**
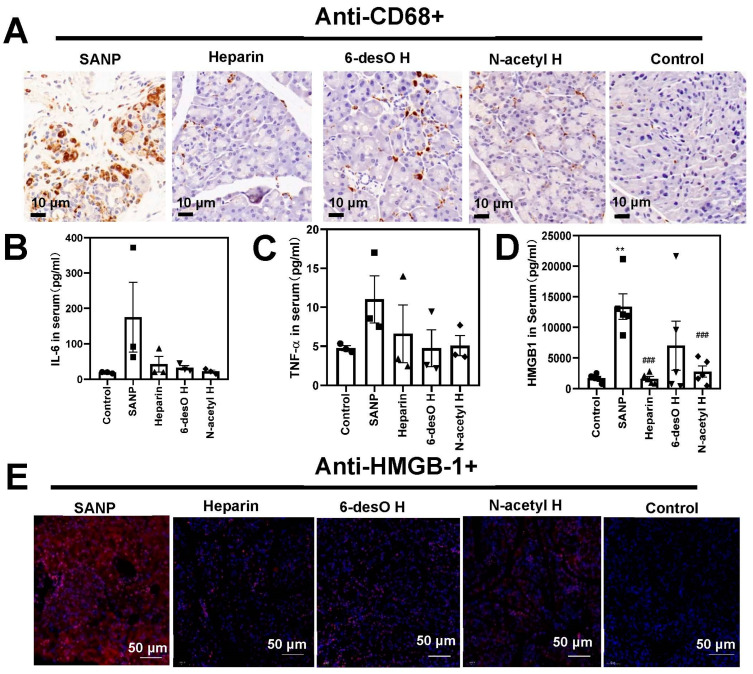
Heparin alleviated the high inflammatory conditions of SAP by reducing extracellular HMGB-1 levels. (**A**) The macrophages in the pancreas were labeled with anti-CD-68+ by immunochemical staining method. (**B**–**D**) The pro-inflammatory factors including TNF-α (B), IL-6 (**C**) and HMGB-1 (**D**) were analyzed in the serum by ELISA. (**E**) Extracellular HMGB-1 distribution assay in the pancreas by immunochemical staining (red for HMGB-1, and blue for DAPI; ### *p* < 0.001, SAP vs. other groups; ** *p* < 0.01, SAP vs. control groups).

**Figure 4 polymers-14-02470-f004:**
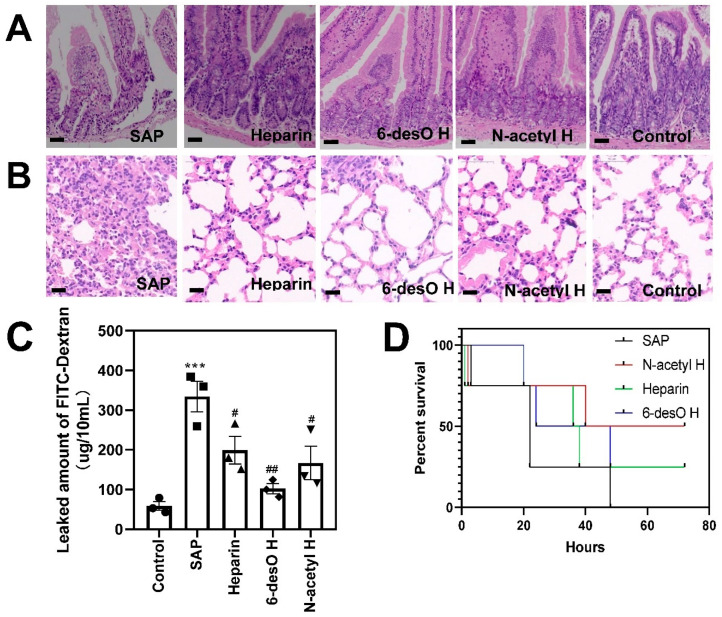
Heparin and its derivatives rescued the SAP mice from multiple organ failure during SAP development. (**A**,**B**) The representative H&E images of intestinal tissues (**A**) and lung (**B**). (**C**) The permeability evaluation of intestinal tissue (Scale Bars = 20 μm). (**D**) The survival rate calculation during 80 h after L-arginine challenges (# *p* < 0.05, ## *p* < 0.01, SAP vs other groups; *** *p* < 0.001, SAP vs. control groups).

**Figure 5 polymers-14-02470-f005:**
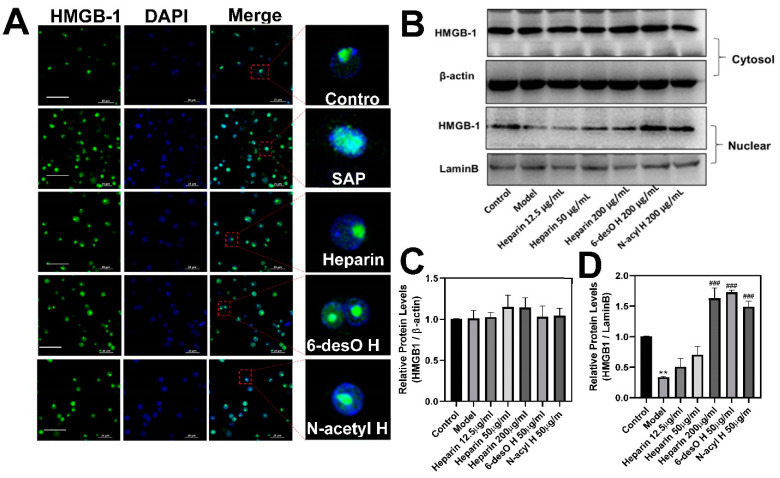
Heparin inhibited HMGB-1 secretion from activated RAW264.7 cell. (**A**) The HMGB-1 distribution in the cells was observed by immunofluorescence (green). The nucleus of the cells was labeled with DAPI (blue). The cells without treatment were regarded as the control group, whereas the SAP and drug groups were activated with the supernatant of the damaged pancreatic tissue of mice for 2 h in advance (Scale Bars = 20 μm). (**B**) The representative images of Western-blotting against HMGB-1 in the cytosol zone and nuclear zone of the cell. (**C**) The relative amount of HMGB-1 in the cytosol zone. (**D**) The relative amount of HMGB-1 in the nuclear zone (### *p* < 0.001, SAP vs. other groups; ** *p* < 0.01, SAP vs. control groups).

**Figure 6 polymers-14-02470-f006:**
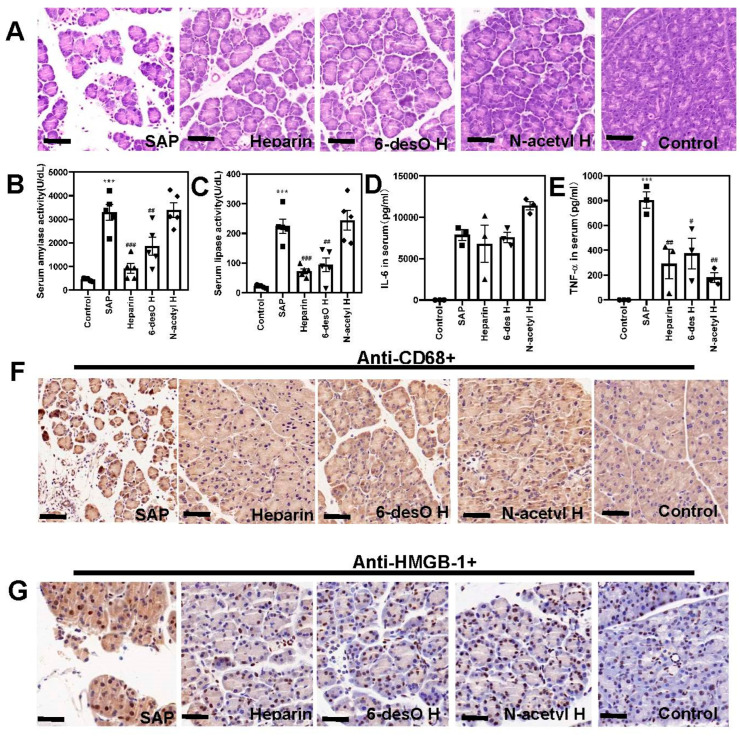
Heparin alleviated caerulein and LPS induced SAP. (**A**) The representative images for the H&E staining of the pancreas. (**B**) The serum α-amylase activity analysis. (**C**) The serum lipase activity analysis. (**D**,**E**) Determination of the TNF-α (**D**), and IL-6 (**E**) in the serum. (**F**) Analysis of the macrophage infiltration of the pancreas by immunohistochemistry (anti-CD68+). (**G**) Pancreatic HMGB-1 distribution by immunohistochemistry (anti-HMGB-1+) (Scale bar=20 μm. # *p* < 0.05, ## *p* < 0.01, ### *p* < 0.001, the SAP group vs. other groups; *** *p* < 0.001, SAP vs. control groups).

## Data Availability

Data available on request from the authors.

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
