# Peer review of "Heparin Protects Severe Acute Pancreatitis by Inhibiting HMGB-1 Active Secretion from Macrophages"

_polymers, 2022, doi:10.3390/polym14122470_

Round 1

Reviewer 1 Report

In this manuscript, the authors performed Heparin Protects the Severe Acute Pancreatitis by Inhibiting HMGB-1 Active Secretion from Macrophage. In my opinion, some issues should be further addressed and I hope the following comments could be helpful for improving their paper.

1. Kindly draw graphical abstract and insert it onto the main manuscript to draw readers' attention/

2. Abstract is poorly written, kindly add the main results as well in the manuscript.

3. What is the difference b/w Histopathology and TUNEL staining?

4. Why Heparin exerted a better effect compared with the N-acetyl H and 6-desO H group.?

5. Why did Acute lung injury occur in the SAP group by histological observation?

6. Discussion: This part requires a thorough development. The authors should clarify the signalized doubts. They should demonstrate the advantages and disadvantages of the proposed system against the background of similar systems described earlier.

7.  Please revisit the entire manuscript for minor grammar issues. The writing although good needs to be corrected for grammar and sentence construction. I also highly recommend the authors to streamline their writing to keep the underlying conclusions precise and clear. The transitions between ideas seem disconnected. These would only help the reader get more from the review and improve on its quality and appeal

Author Response

In this manuscript, the authors performed Heparin Protects the Severe Acute Pancreatitis by Inhibiting HMGB-1 Active Secretion from Macrophage. In my opinion, some issues should be further addressed and I hope the following comments could be helpful for improving their paper.

1. Kindly draw graphical abstract and insert it onto the main manuscript to draw readers' attention/
Answer:  Thank you for your valuable suggestions. The graphical abstract has been inserted into the main manuscript.

2. Abstract is poorly written, kindly add the main results as well in the manuscript.
Answer: The abstract has been re-written according to your suggestions.

3. What is the difference b/w Histopathology and TUNEL staining?
Answer: TUNEL staining mainly reflect the apoptosis of the cells. Whereas histopathology (H&E staining) is a common method to observe the cell morphology. Both necrosis and apoptosis could be observed with this method. The two methods were both used to investigate the damage degree of the cells, and the results support each other. We have replaced histopathology with H&E staining.

4. Why Heparin exerted a better effect compared with the N-acetyl H and 6-desO H group?
Answer: The biological functions of heparin are believed to be dependent on the interaction of heparin with key protein, which have a heparin binding domain in common. The charge density of heparin contribute largely to the affinity between the bio-macromolecules. The interactions for N-acetyl H and 6-desO H were weakened, because the charge density was reduced after partially desulfation. As a result, heparin exerted a better effect compared with the N-acetyl H and 6-desO H groups. The description has been added in the manuscript along with the following reference. (Weinhart M, Gröger D, Enders S, et al. The role of dimension in multivalent binding events: structure–activity relationship of dendritic polyglycerol sulfate binding to L‐selectin in correlation with size and surface charge density[J]. Macromolecular bioscience, 2011, 11(8): 1088-1098.)

5. Why did Acute lung injury occur in the SAP group by histological observation?
Answer: SAP can easily lead to systemic inflammatory response syndrome (SIRS) and multiple organ dysfunction syndromes (MODS). Acute lung injury (ALI) is one of the most serious complications of SAP. See the following references (Ge P, Luo Y, Okoye C S, et al. Intestinal barrier damage, systemic inflammatory response syndrome, and acute lung injury: A troublesome trio for acute pancreatitis[J]. Biomedicine & Pharmacotherapy, 2020, 132: 110770.) The description has been added in the manuscript.

6. Discussion: This part requires a thorough development. The authors should clarify the signalized doubts. They should demonstrate the advantages and disadvantages of the proposed system against the background of similar systems described earlier.
Answer: The discussion has been modified according to your suggestion.

7. Please revisit the entire manuscript for minor grammar issues. The writing although good needs to be corrected for grammar and sentence construction. I also highly recommend the authors to streamline their writing to keep the underlying conclusions precise and clear. The transitions between ideas seem disconnected. These would only help the reader get more from the review and improve on its quality and appeal
Answer: Thank you for your valuable suggestions. We have modified the language with MDPI service. We have also tried our best to streamline the writings in the revised manuscript. We appreciated any further suggestions.

Reviewer 2 Report

It is acknowledged that LMWH could improve the prognosis of SAP and has a potential role in reducing hospital stay, mortality, incidences of multiple organ failure, pancreatic pseudocyst, and operation rate (J Dig Dis. 2019;20(10):512-522). In addition, in patients and animals with AP, serum levels of HMGB1 are significantly increased and positively correlate with the severity of the disease.

In this manuscript, Yang J et al. display that heparin and the non-anticoagulant derivatives had the therapeutic potential for preventing the SAP progression by inhibiting HMGB-1 active secretion from macrophages.

These findings are of great interest in better understanding the pathogenetic mechanisms of AP and their implications appear essential in the development of new therapeutic protocols.

The research design is appropriate, the methods adequately described, and the results clearly presented so, I have no particular suggestions. Of interest, recently, Kang R et al. (Gastroenterology. 2014;146(4):1097-1107) in 2 mouse models of acute pancreatitis showed that intracellular HMGB1, on the contrary to extracellular HMGB1, appears to prevent nuclear catastrophe and release of inflammatory nucleosomes to block inflammation. I suggest the authors comment on this study in the discussion.

Author Response

reviewer 2:
It is acknowledged that LMWH could improve the prognosis of SAP and has a potential role in reducing hospital stay, mortality, incidences of multiple organ failure, pancreatic pseudocyst, and operation rate (J Dig Dis. 2019;20(10):512-522). In addition, in patients and animals with AP, serum levels of HMGB1 are significantly increased and positively correlate with the severity of the disease.

In this manuscript, Yang J et al. display that heparin and the non-anticoagulant derivatives had the therapeutic potential for preventing the SAP progression by inhibiting HMGB-1 active secretion from macrophages.

These findings are of great interest in better understanding the pathogenetic mechanisms of AP and their implications appear essential in the development of new therapeutic protocols.

The research design is appropriate, the methods adequately described, and the results clearly presented so, I have no particular suggestions. Of interest, recently, Kang R et al. (Gastroenterology. 2014;146(4):1097-1107) in 2 mouse models of acute pancreatitis showed that intracellular HMGB1, on the contrary to extracellular HMGB1, appears to prevent nuclear catastrophe and release of inflammatory nucleosomes to block inflammation. I suggest the authors comment on this study in the discussion.

Answer: Thank you for your valuable suggestions. We have added the reference in the discussion section.
